# Tuberculosis among People Living on the Street and Using Alcohol, Tobacco, and Illegal Drugs: Analysis of Territories in Extreme Vulnerability and Trends in Southern Brazil

**DOI:** 10.3390/ijerph19137721

**Published:** 2022-06-23

**Authors:** Alessandro Rolim Scholze, Josilene Dália Alves, Thaís Zamboni Berra, Antônio Carlos Vieira Ramos, Flávia Meneguetti Pieri, Sandra Cristina Pillon, Júlia Trevisan Martins, Maria José Quina Galdino, Emiliana Cristina Melo, Felipe Mendes Delpino, Ariela Fehr Tártaro, Inês Fronteira, Ricardo Alexandre Arcêncio

**Affiliations:** 1Department of Maternal-Infant and Public Health Nursing, Ribeirão Preto College of Nursing, University of São Paulo, Ribeirão Preto 14040-902, Brazil; thais.berra@usp.br (T.Z.B.); antonio.ramos@usp.br (A.C.V.R.); felipedelpino@usp.br (F.M.D.); ariela.fehr@gmail.com (A.F.T.); ricardo@eerp.usp.br (R.A.A.); 2Institute of Biological Sciences and Health, Federal University of Mato Grosso, Barra do Garças 78605-091, Brazil; josilene.alves@ufmt.br; 3Department of Nursing, State University of Londrina, Londrina 86057-970, Brazil; fpieri@uel.br (F.M.P.); jtmartins@uel.br (J.T.M.); 4Department of Psychiatric Nursing and Human Sciences, Ribeirão Preto College of Nursing, University of São Paulo, Ribeirão Preto 14040-902, Brazil; pillon@eerp.usp.br; 5Department of Nursing, State University of Northern Paraná, Bandeirantes 86360-000, Brazil; mariagaldino@uenp.edu.br (M.J.Q.G.); ecmelo@uenp.edu.br (E.C.M.); 6Global Health and Tropical Medicine, Instituto de Higiene e Medicina Tropical, University Nova de Lisboa, 1349-008 Lisboa, Portugal; ifronteira@ihmt.unl.pt

**Keywords:** tuberculosis, vulnerable populations, homeless persons, substance-related disorders

## Abstract

(1) Background: Tuberculosis presents an epidemiological trend toward inequality, especially among people in social exclusion and situations of vulnerability. This study aimed to analyze territories with a concentration of people diagnosed with tuberculosis in a street situation and who partake in chronic use of alcohol, tobacco, and illicit drugs. We also analyzed trends in this health condition in southern Brazil. (2) Methods: Ecological study, developed in the 399 municipalities of Paraná, southern Brazil, with all tuberculosis cases in the homeless population registered in the Information System of Notifiable Diseases between 2014 and 2018. For data analysis, we used descriptive statistics, the Prais–Winsten autoregression method for the time series, and the Getis-Ord Gi technique* for spatial analysis. (3) Results: in total, 560 cases were reported. We found a predominance of alcohol, smoking, and illicit drug users, with an increasing trend in the state and clusters of spatial risk in the East health macro-region. (4) Conclusions: We observed territories with critical levels of highly vulnerable people who use psychoactive substances and are in a street situation. The results highlight the importance of incorporating public policies of social protection for these individuals and resolutive health services that receive these cases and assist in eradicating TB.

## 1. Introduction

Tuberculosis (TB) is an infectious disease caused by *Mycobacterium tuberculosis*. Considered a serious public health problem, it is among the ten most common causes of preventable death in the world [1,2,3]. Historically, TB presents an epidemiological trend toward inequality, especially among people in social exclusion and situations of vulnerability [4]. Thus, the homeless population, the population deprived of liberty (PPL), people infected with human immunodeficiency virus (HIV), and users or addicts of psychoactive substances, alcohol, tobacco, and other drugs [5,6] are considered at high risk of contributing to the spread of this disease.

In this sense, the homeless population is defined as a heterogeneous population that has extreme poverty, interrupted or weakened family ties, and the absence of regular conventional housing in common. This population uses public spaces and degraded areas as a living and subsistence space, temporarily or permanently, as well as reception units for temporary accommodation or as temporary housing [7].

Being homeless is an aggravating factor for the spread of TB in social exclusion groups since TB infection is transmitted via the respiratory route by inhaling the sputum droplets expelled by an infected person [6]. Given the social situation of homeless people, there may be a considerable spread of the disease among them since they tend to congregate in groups. Especially among the homeless population, TB has a high prevalence due to close contact within a large contingent of homeless people and frequent movement between different shelters and territories [8].

In addition, most infected people do not present signs and symptoms of the disease and are classified as carriers of latent infection by TB. The absence of signs and symptoms prevents the search for treatment and intensifies the spread of the disease. It is estimated that approximately 2 to 3 billion people in the world are infected with TB, and of these, about 5% to 15% will evolve to active TB during their lifetime [9,10]. Studies point out that TB among the homeless population is the third-largest cause of illness and has a 10 to 85 times higher rate of developing into latent or active TB infection when compared with the general population [11]. In the United States, about 5% of TB cases have been homeless at some point in the 12 months prior to diagnosis [12].

Among individuals diagnosed with TB in the United States, the incidence rate is 36 to 47 cases per 100 inhabitants, compared to the incidence in the general population of 2.8 cases per 100 inhabitants [13]. In Germany, TB in the homeless has 30 cases per 100 inhabitants, compared to the general population of 18 cases per 100 inhabitants [8]. In western Europe, the prevalence of TB among homeless people ranges from 1% to 2% for active TB infection and up to 45% for latent infection [14].

TB among the homeless population is a challenge since even those countries with low endemicity show a high TB incidence among homeless people. This factor contributes to the maintenance of the TB burden in these countries, thus making it difficult to achieve the goals established by the End TB strategy.

This evidence indicates that vulnerable populations have become a major challenge for countries to develop strategies and achieve their TB elimination goals globally. In social epidemiology, there are several resources available that demonstrate the strength of the social determinants of territories in the TB progression chain, with a large number of studies using spatial analysis in the general population. However, this is not the reality for the homeless population since an address is an aspect required for geoprocessing/georeferencing; as such, many studies using these approaches exclude this population. 

Another gap is that vulnerability is multifaceted. Within the same context, many vulnerabilities end up countering each other, causing cases to become overly complex, requiring equally complex approaches toward alleviating suffering and bringing solutions to the problem. Given the presented problems, the present study is of great importance. It brings together the question of territories and the homeless population with TB and the relationship with alcohol, tobacco, and illicit drugs; as a result, disease control is far from being achieved in some regions and territories. Therefore, this study aimed to analyze territories where there is a concentration of people diagnosed with tuberculosis, in a street situation, and who partake in chronic use of alcohol, tobacco, and illicit drugs. We also analyzed trends in this health condition in southern Brazil.

## 2. Materials and Methods

### 2.1. Study Design

Ecological study [15].

### 2.2. Study Location and Population

This study was carried out in the 399 municipalities of Paraná, located in the southern region of Brazil at the geographic coordinates 24°59′ S latitude and 53°56′ W longitude; the estimated population is 11.34 million inhabitants. It is the fifteenth state of Brazil with the largest national territory and the fifth-highest population [5].

In 2020, Paraná had 2,190 TB cases, obtaining an incidence of 19 cases per 100,000 inhabitants; there were 157 deaths, with an incidence of 1.4 per 100,000 inhabitants. About 6.2% of patients abandon treatment [3].

According to the Inter-sector Committee for Monitoring and Monitoring the Population Policy in Street Situation (CIAMP Street/PR), which aims to enable and assist in the implementation and monitoring of public policies aimed at the population in a street situation, it is estimated that, in 2021, the state had about 8,659 people registered at CIAMP street/PR. It is noteworthy that this may be an under notification, indicating that the number should be higher, mainly due to the current situation in the country, as well as the high turnover of homeless people. Between 2015 and 2020, the state showed a 65% increase in TB cases in the homeless population, and in 2020, a 6.8% increase in the percentage of cases [16].

To develop the spatial distribution, we subdivided the 399 municipalities that make up the state of Paraná into four health macro-regions (East, North, Northwest, and West) and used these as a unit of geographical analysis. Figure 1 illustrates the location of the state and its health macro-regions.

### 2.3. Inclusion Criteria

We considered all cases of TB in the homeless population obtained in the Information System of Notifiable Diseases (SINAN) from 2014 to 2018. These data were made available by the Health Department of the State of Paraná in an electronic spreadsheet in Excel. 

The adopted inclusion criteria were people diagnosed with TB, included in the notification form as a person living on the street, alcohol, tobacco, or illicit drug user, and aged 18 years or older. To identify risk areas and time series, the population was subdivided into general TB, TB alcoholism, TB smoking, and TB users of illicit drugs. It is emphasized that an individual may belong simultaneously to one or more study groups.

### 2.4. Analysis Plan

First, the descriptive statistics of the data were performed utilizing absolute and relative frequency, which used the following variables: alcoholism, smoking, and illicit drugs, sex, age group, race/color and education, TB/HIV coinfection, diabetes mellitus, mental disorder, and data related to the clinical profile of TB, such as type of entry, form, radiography, sputum smear, histopathology, molecular test, and case closure. It should be noted that the terminologies described in the Sinan notification form were kept in full; it should be noted that this form is a structured document recommended by the Ministry of Health and applies throughout the national territory of Brazil.

The IBM SPSS software version 25 was used to analyze the data.

### 2.5. Time Series

Time series are characterized as observations taken sequentially over time [17]. It is worth noting that the temporal trend refers to the direction in which the time series develops according to a determined time interval, which may follow a growth, decrease, or stationary pattern [18,19]. The month and year of notification of TB cases were used to perform the analysis.

The Prais–Winsten autoregression method [18] was performed in STATA software, version 14, to classify the event’s temporal trend as increasing, decreasing, or stationary in the study period. When the temporal trend was classified as increasing or decreasing, the percentage of monthly variation (MPC—monthly percent change) and their respective 95% confidence intervals (95 CI) were calculated [17].

Next, we used the robust time series decomposition method called seasonal trend by Loess (STL by Loess) [16]. This decomposition method is based on a locally weighted regression (Loess). It is the method used to estimate nonlinear relations, managing to separate the components that make up a time series: trend, seasonality, and noise [19,20]. For the analysis, RStudio software and the forecast package were used.

In contrast to the Prais–Winsten method, where the time trend assessment is global and a constant is generated by classifying the whole period under study, the STL method allows for the evaluation of the time trend over the whole period under analysis, notes variations over time, checks if the trend has always been increasing/decreasing/stationary, or if there have been periods of variations with peaks and/or decreases.

### 2.6. Spatial Analysis

The techniques called Getis-Ord General G and Getis-Ord Gi* were used to identify whether clusters were formed, using the number of cases per municipality. The Getis-Ord General G technique is based on the Moran Global Index, and, as in inferential statistics, the results are based on the null hypothesis that there is no spatial grouping. If p is significant, the null hypothesis can be rejected, and the z-score value becomes important, where its values of ±3 represent a 99% confidence level [21,22]. If the z-score value is positive, the observed G-Index is higher than expected, indicating high event indices grouped in the area under study. A negative z-score value of the G-index is considered lower than the expected index, indicating that low values are grouped in the study area [23].

The Getis-Ord Gi* technique indicates a local association, considering the values for each census tract from a neighborhood matrix. In this analysis, a z-score was generated for statistically significant municipalities, and the higher the z-score, the more intense the grouping of high values is (hotspot). The logic is the same for the z-negative score, i.e., the lower the z-score, the more intense the grouping of low values (cold spot) [23].

In addition to the z-score, the *p*-value and significance level (Gi-Bin) are also provided, which identify statistically significant hot and cold spots. The values may vary between +/− 3 and reflect statistical significance with a confidence level of 99%, +/− 2 with a confidence level of 95%, and +/− 1 with a confidence level of 90%, with a zero-value corresponding to non-statistically significant areas [23].

### 2.7. Ethical Aspects

The study was approved by the Research Ethics Committee of the *Escola de Enfermagem de Ribeirão Preto da Universidade de São Paulo*, under CAAE (Process number: 24963319.1.0000.5393).

## 3. Results

Between 2014 and 2018, 560 cases of TB were reported among the homeless population. The analysis showed that the consumption of psychoactive substances was prevalent in homeless users (*n* = 420; 36.30%), as were smokers (*n* = 382; 33.02%), and users of other drugs (*n* = 355; 30.68%).

Table 1 shows the characterization of TB cases in the homeless population subdivided by type of psychoactive substance. The total sample showed a higher predominance of TB in males, in the age group above 40 years of white race/color, with education from the fifth to eighth grade of elementary school, and living in the urban perimeter. The sociodemographic characteristics of the homeless population of users of alcohol, tobacco, and illicit drugs are similar to that of the general homeless population. Some peculiarities were noted, such as one-third of alcohol and tobacco users had a poor education (first to fourth grade), 43.1% (*n* = 153) of other drug users were younger (30 to 39 years of age) when compared to alcohol users (50.2%; *n* = 211) and tobacco (47.1%; *n* = 180) users, who were in the age group of 40 years or older. Notably, 73.7% (*n* = 412) of the general street population who had TB did not receive government benefits. Regarding the presence of chronic diseases, it was observed that TB-AIDS coinfection was the most prevalent among all variables studied, followed by the categories “other diseases” and mental illness.

Regarding the clinical profile of TB cases in the homeless population (Table 2), in the general street population, there was a prevalence of new cases, pulmonary TB, with suspect chest radiography, positive sputum culture, histopathology not performed, HIV negative test, molecular tests not performed, sensitivity tests not reported, and curative TB treatment. When analyzing the clinical profile of cases associated with the type of drug, it was found that the same specificities of the total population occurred, noting that, regarding the closure situation, abandonment and death due to TB and other causes were more present among patients who used alcohol, tobacco, and other drugs.

The temporal trend of TB cases in the homeless population (Table 3) presented an increasing scenario for all categories analyzed. For the population using alcohol, the growth was 29.71% per month (95% CI 18.03–42.56), tobacco 27.93% per month (95% CI 16.68–39.95), other drugs 30.31% per month (95% CI 18.57–43.21), and general drugs 38.35% (95% CI 23.31–55.23).

Figure 2 shows the time series decomposition technique. We found an increase in the temporal trend of TB cases in the homeless population for alcohol, tobacco, and other drugs and in the general street population. We identified slight fluctuations of the temporal trend concerning the number of cases over the months, thus corroborating the findings presented in Table 1 referring to the Prais–Winsten analysis.

For the results of the global spatial association (G), we observed that the values of the z-score and the pseudo-significance test confirmed the non-randomness of TB cases in the population using alcohol (z-score of 3.00 and *p* <0.00), tobacco (z-score 3.12 and *p* < 0.00), illicit drugs (z-score 2.79 and *p* < 0.00), and general drugs (total) (z-score 3.78 and *p* < 0.00). Figure 3 shows the local spatial association (Gi*) of TB cases, which shows hotspots in the Eastern macro-region in the metropolitan region of Curitiba for users of alcohol, tobacco, and illicit drugs and in the general street population.

## 4. Discussion

We analyzed territories with a concentration of people diagnosed with tuberculosis, in a homeless situation, and who partake in the chronic use of alcohol, tobacco, and illicit drugs; we also assessed the trends of this health condition in southern Brazil. We observed that, in the south of Brazil, despite being considered one of the most developed regions of the country, there were still clusters or areas of the population with extreme vulnerability, people diagnosed with TB, using psychoactive substances, and still on the street. There was also a growing increase in this situation in the East health macro-region (metropolitan region of Curitiba, the state capital).

International and national studies indicate that the homeless population has a high prevalence of TB associated with the use of alcohol, tobacco, and illicit drugs, which are risk factors that favor the development of new cases and the maintenance of the TB transmission cycle [4,8,24]. Thus, the homeless population has a 48 to 67 times greater chance of developing TB when compared to the general population [25].

A study conducted in the United States between 2006 and 2010 found that the annual incidence of TB among the homeless population ranged from 36 to 47 cases per 100,000 inhabitants [26]. In Germany, the incidence of TB is 30 cases per 100,00 inhabitants [8], whereas, in Brazil, there are no specific data that address this population.

It is worth noting that Brazil does not have official data on the size of the homeless population, considering that the demographic census does not include homeless residents in the investigation. Since these data are collected in fixed residences, it is impossible to correctly quantify these individuals, which makes them invisible [27].

The high prevalence of TB in this population is related to precarious urbanization, sanitary conditions (close contact with a contingent of people and movement between different shelters), social exclusion, and extreme poverty [28]. These individuals experience a highly vulnerable environment on a daily basis, which leads to health problems and infections, including TB [8]. This population also has high rates of avoidable death from all causes compared to the general population, which exemplifies the influence of inequalities and social determinants of health in the illness process of these individuals [29].

In addition, this population has specificities that favor the development of health problems, such as poor education, which consequently generates ignorance regarding primary health care and the health-disease process. Moreover, many have difficulties in accessing health services, experience stigma, prejudice, conflicting or nonexistent family ties, and the lack of a life project, among other factors [4,28].

The homeless population presents a deficit in self-care concerning hygiene and eating habits, which contributes to a precarious lifestyle and a high prevalence of complications due to chronic diseases, infectious conditions, and injuries related to violence [30]. Therefore, strategic actions should be implemented for the coverage and supply of health services, social assistance, and effective housing to these individuals so that basic human needs are guaranteed.

When evaluating the spatial determinants and the conformation of the clusters when applying the Getis-Ord Gi* technique, clusters of hotspots were identified in the East health macro-region. It is estimated that, in Brazil, there is a population of approximately 100,000 homeless and that 75% of this population lives in municipalities with more than 100,000 inhabitants [27].

Access to housing is understood as an important determinant of health, and homelessness is directly related to an increase in morbidity and mortality compared to the sheltered population. Mortality among the homeless population is 3 to 11 times higher [31].

In addition, it can be highlighted that mobility among homeless people, loss of follow-up, lack of attendance to follow-up appointments, unstable housing, incarceration, fear of invasive investigations and the side effects of TB treatment, transportation and location of health services, lack of flexible hours of care, and problems with remembering commitments and the correct use of medicines are some barriers that hinder a favorable outcome of TB treatment among the homeless population and users/addicts of psychoactive substances [32].

It is necessary to recognize the magnitude of the public health problem present in large cities among the street population and to implement public policies aimed at this heterogeneous group that is in extreme social vulnerability [33,34]. Thus, implementing policies in terms of social support and reintegration of the street population into society is extremely important, as are programs aimed at income distribution, such as the *Bolsa Família* Program. The Bolsa Família Program aims to provide income to families in poverty and extreme poverty. The aid seeks to overcome the situation of vulnerability and poverty, thus ensuring the right to food and access to education and health [35]. The implementation of the Bolsa Família Program provided an increase in the rate of adherence and cure of patients with TB, highlighting the need to implement policies aimed at social support [36].

Another behavior analyzed in this study was that TB is growing among alcohol, tobacco, and illicit drug users. This result is extremely important since the higher the consumption of psychoactive substances by the homeless population, the worse the prognosis of these individuals; other problems include maintenance, increased incidence, and the recurrence of new cases of TB and drug-resistant TB TB-DR. In addition, homeless people who use psychoactive substances move through spaces/territories with a high number of individuals, a characteristic that contributes to a greater spread of the disease.

Therefore, the use of psychoactive substances is considered an aggravating and complicating factor for the containment and eradication of TB in the world. It is noteworthy that the Sustainable Development Goals of the World Health Organization reinforce the importance of implementing prevention and treatment of disorders related to the use of alcohol and other drugs among vulnerable populations [37].

In this sense, investing in public policies, health services targeted at vulnerable populations, and skilled professionals can be an effective tool for early diagnosis, correct treatment, a decrease in the incidence of TB, guaranteed access to health services and social support, and a direction for the social reintegration of individuals who are on the street.

Moreover, the use of psychoactive substances among vulnerable populations promotes community transmission and TB contamination since these individuals live in an environment of extreme sanitary prevarication and share objects and materials such as pipes and needles to make use of psychoactive substances. This characteristic contributes to the spread and dissemination of latent and active TB in these territories [38].

It was also observed that the worst outcomes in treatment and death were present in vulnerable populations that used or were dependent on any substance. Therefore, the vulnerable population has a high TB infectivity rate and thus makes it a strategic population for actions aimed at eradicating TB in the world. Developing public policies aimed at vulnerable populations is a way to contain the spread of the disease in these territories [33,39] and thus achieve the goals established through the End TB strategy.

In order to reduce the infection rate and decrease the number of new cases of TB or relapses, countries need to develop specific strategies aimed at vulnerable populations, such as users of alcohol, tobacco, and other drugs, whereas by implementing health actions directed at these individuals, it is possible to ensure early detection and diagnosis of TB and, consequently, effective treatment [39]. It should be emphasized that countries need to act and invest in research to highlight the territories with the greatest risk of illness and thus identify why people get sick in different territorial regions [40].

Ensuring health services that are responsive and targeted to vulnerable populations is of utmost importance, as is ensuring reception and, consequently, providing the correct sequence during treatment. However, what is observed in health services is a barrier to access and health care for vulnerable populations, which makes these people invisible to the public health system and contributes to the maintenance of TB and other health problems [41].

Strategies that can enhance and provide more comprehensive health care to these populations are the incorporation of health services within vulnerable territories and the implementation of street offices and health care within prisons. The purpose of street offices is to expand the access of the homeless population to health services and to provide comprehensive health care to individuals who are in vulnerable conditions or with fragile family ties [41,42].

The street office has a work dynamic that allows for the active search for people living on the street in several places in the city, valuing welcoming and the creation of links in order to supply the needs of these individuals, without judgment or social standards, and guaranteeing them the right to health advocated in the constitution [43].

It is also relevant to consider the implementation in the country of networks that seek better social protection for vulnerable populations, considering that, when investing in measures that aid equity in health and poverty reduction, a reversal will occur in the context of morbidity and mortality of TB. This strategy is emphasized and recommended by the World Health Organization.

Another measure to be adopted is the incorporation of health policies aimed at reducing the consumption of psychoactive substances in vulnerable territories and throughout society. One aim is the development of prevention and promotion actions related to early consumption among children and adolescents, promoting actions in vulnerable populations, and, in general, a conscious consumption of the substance; by incorporating measures aimed at conscious consumption, this factor reduces exposure to risk, as well as health problems [16].

When analyzing the strategies used to control TB in other countries, it was found that, in the United States, actions such as monetary incentives and education programs are implemented to improve adherence to treatment since these measures help in adherence to treatment and contribute to a better level of treatment completion and thus interruption of the TB transmission cycle [44].

Another feature that helps to reduce the risk of infection and improves the chance of successful treatment is the implementation of programs focused on the health education of the homeless population. In other words, the development of early interventions aimed at preventing TB infection, promoting early diagnosis, and reducing treatment abandonment, drug-resistant TB, and death.

Finally, this study provides significant contributions and joins the global effort to control TB since it was possible to incorporate statistical methods that identified areas of spatial risk for TB among users dependent on alcohol, tobacco, and illicit drugs.

## 5. Conclusions

This study advances scientific knowledge since few studies have addressed the geoepidemiology of TB between the homeless population and the relation of chronic use of alcohol, tobacco, and illicit drugs. In this sense, it was possible to highlight territories at risk for the development of TB in the homeless population and their relationship with the territories that have a large number of people.

It is necessary to incorporate public policies of social protection for these individuals and resolutive, welcoming health services that will assist in eradicating TB.

## Figures and Tables

**Figure 1 ijerph-19-07721-f001:**
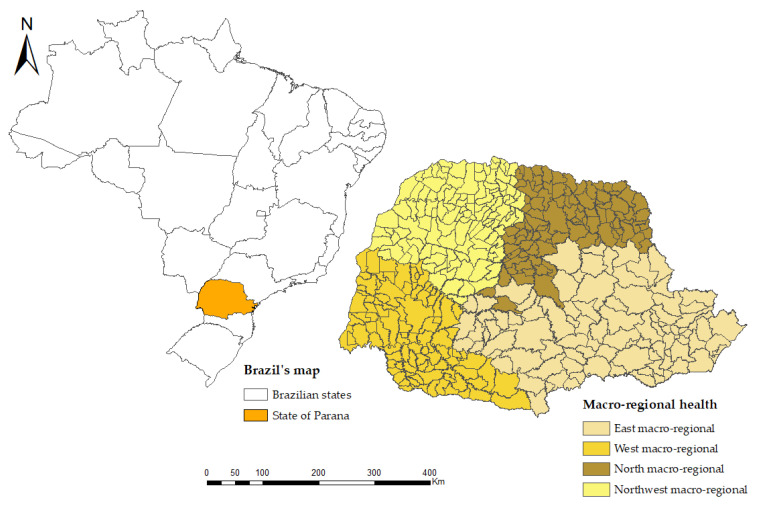
Geographical location of health macro-regions according to municipalities in the state of Paraná.

**Figure 2 ijerph-19-07721-f002:**
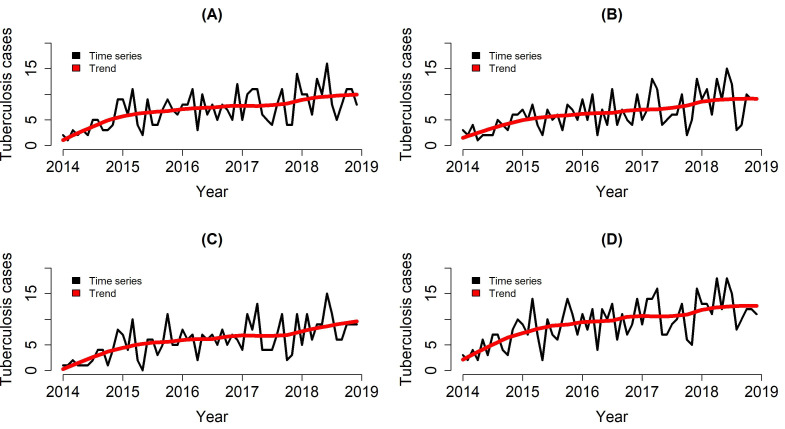
Time series of the homeless population diagnosed with tuberculosis in the state of Paraná, Brazil (2014–2018). (**A**) Alcohol; (**B**) Tobacco; (**C**) Illicit drugs; (**D**) General street population.

**Figure 3 ijerph-19-07721-f003:**
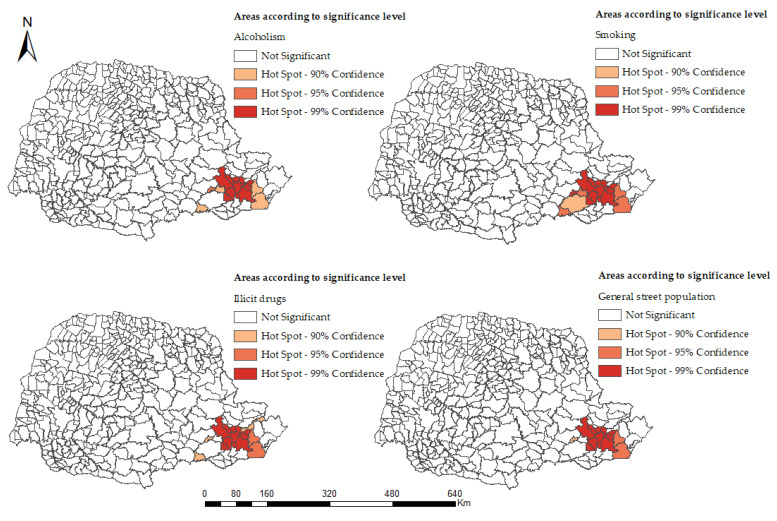
High clusters and low clusters of tuberculosis cases in the homeless population in the state of Paraná, Brazil (2014–2018).

**Table 1 ijerph-19-07721-t001:** Sociodemographic characterization of tuberculosis cases in the homeless population of the state of Paraná, Brazil (2014–2018).

Variables	Street Population Using Alcohol		Street Population Using Tobacco		Street Population Using Illicit Drugs		General Street Population	
	*n*	%	*n*	%	*n*	%	*n*	%
Gender								
Male	362	82.3	318	83.2	281	79.2	472	84.4
Female	58	13.2	64	16.8	74	20.8	87	15.6
Age group								
18 to 29 years	61	14.5	65	17.0	77	21.7	97	17.4
30 to 39 years	148	35.2	137	35.9	153	43.1	197	35.2
40 or more	211	50.2	180	47.1	125	35.2	265	47.4
Race								
White	217	51.7	197	51.6	180	50.7	292	52.2
Black	51	12.1	47	12.3	39	11.0	61	10.9
Yellow	2	0.5	1	0.3	4	1.1	4	0.7
Brown	143	34.0	125	32.7	122	34.4	183	32.7
Indigenous	1	0.2	1	0.3	1	0.3	2	0.4
Schooling								
Illiterate	11	2.6	11	2.9	11	3.1	17	3.0
First to fourth grade	126	30.0	111	29.1	84	23.7	149	26.7
Fifth to eighth grade	112	26.7	103	27.0	108	30.4	157	28.1
> Eight years of study	39	9.3	41	10.7	35	9.9	55	9.8
Housing perimeter								
Urban	367	87.4	334	87.4	323	91.0	485	86.8
Rural	16	3.8	16	4.2	10	2.8	25	4.5
Periurban	1	0.2	-	-	-	-	2	0.4
Government benefit								
Yes	22	5.2	24	6.3	24	6.8	39	7.0
No	312	74.3	288	75.4	266	74.9	412	73.7
Alcoholism								
Yes	420	100.0	318	83.2	284	80.0	420	75.1
No	-	-	56	14.7	60	16.9	118	21.1
Smoking								
Yes	318	75.7	382	100.0	274	77.2	382	68.3
No	82	19.5	-	-	65	18.3	140	25.0
Other drugs								
Yes	284	67.6	274	71.7	355	100.0	355	63.5
No	110	26.2	94	24.6	-	-	163	29.2
HIV/AIDS								
Yes	110	26.2	106	27.7	117	33.0	153	27.4
No	291	69.3	258	67.5	224	63.1	378	67.6
Diabetes mellitus								
Yes	15	3.6	14	3.7	11	3.1	19	3.4
No	389	92.6	354	92.7	331	93.2	511	91.4
Mental illness								
Yes	28	6.7	28	7.3	26	7.3	36	6.4
No	371	88.3	338	88.5	310	87.3	487	87.1
Other diseases								
Yes	50	11.9	46	12.0	41	11.5	64	11.4
No	287	68.3	263	68.8	245	69.0	389	69.6

**Table 2 ijerph-19-07721-t002:** Clinical profile of tuberculosis cases in the homeless population of the state of Paraná, Brazil (2014–2018).

Variables	Street Population Using Alcohol		Street Population Using Tobacco		Street Population Using Illicit Drugs		General Street Population	
	*n*	%	*n*	%	*n*	%	*n*	%
Type of entry								
New case	256	61.0	232	60.7	217	61.1	351	62.8
Recurrence	37	8.8	33	8.6	31	8.7	46	8.2
Reentry after loss to follow-up	80	19.0	73	19.1	72	20.3	102	18.2
Do not know	2	0.5	-	-	2	0.6	3	0.5
Transfer	41	9.8	41	10.7	31	8.7	51	9.1
Clinical form								
Pulmonary	378	90.0	343	89.8	313	88.2	504	90.2
Extrapulmonary	20	4.8	17	4.5	19	5.4	24	4.3
Pulmonary + extrapulmonary	22	5.2	22	5.8	23	6.5	31	5.5
Chest radiography								
Suspect	356	84.8	329	86.1	292	82.3	466	83.4
Normal	7	1.7	5	1.3	7	2.0	9	1.6
Other pathology	1	0.2	1	0.3	1	0.3	1	0.2
Not performed	52	12.4	43	11.3	52	14.6	78	14.0
Sputum smear microscopy								
Positive	265	63.1	247	64.7	222	62.5	359	64.2
Negative	63	15.0	62	16.2	52	14.6	85	15.2
Not performed	88	21.0	71	18.6	78	22.0	110	19.7
Sputum culture								
Positive	168	40.0	153	40.1	145	40.8	221	39.5
Negative	80	19.0	82	21.5	77	21.7	106	19.0
In progress	15	3.6	14	3.7	11	3.1	18	3.2
Not performed	157	37.4	133	34.8	122	34.4	214	38.3
Histopathology								
Baar positive	46	11.0	45	11.8	41	11.5	67	12.0
Suggestive of TB	12	2.9	10	2.6	9	2.5	14	2.5
Non-suggestive of TB	2	0.5	3	0.8	4	1.1	4	0.7
In progress	12	2.9	9	2.4	6	1.7	13	2.3
Not performed	343	81.7	311	81.4	291	82.0	454	81.2
HIV test								
Positive	113	26.9	107	28.0	120	33.8	156	27.9
Negative	270	64.3	247	64.7	208	58.6	352	63.0
Not performed	36	8.6	27	7.1	27	7.6	50	8.9
Final status								
Cure	152	36.2	149	39.0	114	32.1	200	35.8
Primary abandonment	13	3.1	10	2.6	10	2.8	14	2.5
Abandonment	91	21.7	85	22.3	85	23.9	137	24.5
Death by TB	31	7.4	26	6.8	23	6.5	39	7.0
Death from other causes	44	10.5	36	9.4	39	11.0	52	9.3
Transference	68	16.2	64	16.8	67	18.9	91	16.3
TB-DR	12	2.9	10	2.6	8	2.3	13	2.3
Molecular testing								
Detectable at rifampin	112	26.7	107	28.0	108	30.4	158	28.3
Detectable rifampin resistant	9	2.1	6	1.6	8	2.3	11	2.0
Undetectable	13	3.1	13	3.4	11	3.1	14	2.5
Inconclusive	6	1.4	5	1.3	4	1.1	6	1.1
Not performed	262	62.4	235	61.5	213	60.0	344	61.5
Not reported	18	4.3	16	4.2	11	3.1	26	4.7
Sensitivity test								
Resistant to isoniazid only	6	1.4	6	1.6	3	0.8	7	1.3
Resistance to rifampin only	2	0.5	1	0.3	2	0.6	3	0.5
Resistant to isoniazid and rifampin	4	1.0	4	1.0	2	0.6	5	0.9
Resistant to other first line drugs	3	0.7	4	1.0	4	1.1	4	0.7
Sensitive	74	17.6	66	17.3	62	17.5	98	17.5
In progress	6	1.4	7	1.8	4	1.1	8	1.4
Not performed	61	14.5	51	13.4	45	12.7	77	13.8
Not reported	264	62.9	243	63.6	233	65.6	357	63.9

**Table 3 ijerph-19-07721-t003:** Temporal trend of tuberculosis incidence in the street population according to the consumption of psychoactive substances, Paraná, Brazil (2014–2018).

Variable	Coefficient (CI * 95%)	Temporal Trend	MPC ** (CI95%)
Street population using alcohol	2.47 (1.50–3.54)	Crescent	29.71 (18.03–42.56)
Street population using tobacco	3.32 (1.39–3.32)	Crescent	27.93 (16.68–39.95)
Street population of illicit drugs	2.52 (1.54–3.60)	Crescent	30.31 (18.57–43.21)
General street population	3.19 (1.94–4.60)	Crescent	38.35 (23.31–55.23)

***** Confidence interval ****** Monthly percent change.

## Data Availability

The data presented in this study are available on request from the corresponding author on reasonable request.

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
