# Peer review of "Tuberculosis among People Living on the Street and Using Alcohol, Tobacco, and Illegal Drugs: Analysis of Territories in Extreme Vulnerability and Trends in Southern Brazil"

_ijerph, 2022, doi:10.3390/ijerph19137721_

Round 1

Reviewer 1 Report

Article entitled "Tuberculosis among peoples who were living on the street and used alcohol, tobacco, and illegal drugs: analysis of territories in extreme vulnerability and trends in southern Brazil" authors have discussed the impact of alcohol, tobacco, and illegal drugs: analysis of territories in extreme vulnerability and trends in southern Brazil over tuberculosis. It would be more impactful if they can add TB-HIV coinfection, MDR, and XDR TB among the population.

Overall English editing is needed, Statistical data analysis needs special attention, and Age group/gender wise data needs to include to make the study more impactful.

Author Response

Comments and Suggestions for Authors

Article entitled "Tuberculosis among peoples who were living on the street and used alcohol, tobacco, and illegal drugs: analysis of territories in extreme vulnerability and trends in southern Brazil" authors have discussed the impact of alcohol, tobacco, and illegal drugs: analysis of territories in extreme vulnerability and trends in southern Brazil over tuberculosis. It would be more impactful if they can add TB-HIV coinfection, MDR, and XDR TB among the population.

R - Dear author, considering the study's objective, which is to analyze territories where there is a concentration of people diagnosed with tuberculosis, in a street situation, and who make chronic use of alcohol, tobacco, and illicit drugs. In this sense, the suggestion of including an analysis of TB-HIV, MDR, and XDR TB co-infection among the population will not be possible because including it would go against the initial proposal of this research. However, I thank you for the suggestion, and we will structure a new study proposal considering the inclusion of these variables for the study.

Overall English editing is needed, Statistical data analysis needs special attention, and Age group/gender wise data needs to include to make the study more impactful.

R – The variables age and sex are already included between lines 192 to 194. I ask the reviewer to be more specific about what he would like to be included. However, we have included a paragraph in the discussion about the data related to gender and age.

R – The article has been submitted to native expert reading (Appendix)

Reviewer 2 Report

Authors dare to discuss an interesting topic with good statistics and decent illustration of their work.

More epidemiological evidence should be added in the introduction and compared with the findings presented in the discussion section.

Homeless focused strategies are of interest . Comparison with other countries' strategies has to be further discussed as well.

English editing is needed i.e.

ln 281: " Homeless" instead of "homeless"

ln 344-345 "especially among" should be replaced by "like"

ln 353 "an invisible population" should be replaced by "invisible"

and many other.

The paragraph 371-374 needs to be rephrased.

Author Response

Comments and Suggestions for Authors

Authors dare to discuss an interesting topic with good statistics and decent illustration of their work.

More epidemiological evidence should be added in the introduction and compared with the findings presented in the discussion section.

R – We included a paragraph in which the TB incidence among homeless people was compared with the general population in different countries, including Brazil.

Homeless focused strategies are of interest . Comparison with other countries' strategies has to be further discussed as well.

R – There was the inclusion of strategies adopted in other countries at the end of the discussion.

English editing is needed i.e.

R – The article has been submitted to native expert reading (Appendix)

ln 281: " Homeless" instead of "homeless"

R - changed

ln 344-345 "especially among" should be replaced by "like"

R - changed

ln 353 "an invisible population" should be replaced by "invisible"

R - changed

The paragraph 371-374 needs to be rephrased.

R – The paragraph was restructured as indicated.

Reviewer 3 Report

Scholze et al., presented a manuscript that is right to the point, logical, and concise. The introduction section properly engages readers by describing the challenges and gaps. The methods section provides detailed information on how this study was conducted and the Discussion section clearly address the components of the strategic plan for the future. This is a well-written manuscript overall. My only concern is the Table1 race category. Does these color terms officially used in Brazil? Can they be replaced with other terms? If this is the official classification of race in Brazil, can the authors indicate this somewhere in the method section, since not all countries use this type of color-based race classification?

Author Response

Scholze et al., presented a manuscript that is right to the point, logical, and concise. The introduction section properly engages readers by describing the challenges and gaps. The methods section provides detailed information on how this study was conducted and the Discussion section clearly address the components of the strategic plan for the future. This is a well-written manuscript overall. My only concern is the Table1 race category. Does these color terms officially used in Brazil? Can they be replaced with other terms? If this is the official classification of race in Brazil, can the authors indicate this somewhere in the method section, since not all countries use this type of color-based race classification?

R – We included the justification of race/color in the method and informed that the terminologies described in the Sinan notification form were entirely maintained. In this sense, it uses the description (race/color: white, brown, black, yellow, and indigenous).

Reviewer 4 Report

In this manuscript authors aim to characterize tuberculosis outcome in rural people living on the streets in southern Brazil. Though this study lacks novelty but have significance in terms geographical area, demographics and robust statistical analysis. This is a well written manuscript, I recommend for publication after proofreading minor grammatical corrections. 

Author Response

In this manuscript authors aim to characterize tuberculosis outcome in rural people living on the streets in southern Brazil. Though this study lacks novelty but have significance in terms geographical area, demographics and robust statistical analysis. This is a well written manuscript, I recommend for publication after proofreading minor grammatical corrections

R – The article has been submitted to native expert reading (Appendix)

Round 2

Reviewer 1 Report

Authors have significantly revised the manuscript and now the revised version is in the good form which can be published. 

Reviewer 2 Report

I have no more concerns